# High-frequency variability of $CO_2$ in Grand Passage, Bay of Fundy, Nova Scotia

Rachel M. Horwitz[*1,3], Alex E. Hay[1], William Burt[1,4], Richard Cheel[1], Joseph Salisbury[2], and Helmuth Thomas[1]

[1]Dalhousie University, Department of Oceanography, Halifax, Nova Scotia, Canada
[2]University of New Hampshire, Institute for the Study of Earth, Oceans, and Space, Durham, New Hampshire, USA
[3]*Current affiliation:* Bedford Institute of Oceanography, Fisheries and Oceans Canada, Dartmouth, Nova Scotia, Canada
[4]*Current affiliation:* University of British Columbia, Department of Earth, Ocean and Atmospheric Sciences, Vancouver, British Columbia, Canada

**Correspondence:** Rachel Horwitz (rmhorwitz@gmail.com)

**Abstract.** Assessing changes in the marine carbon cycle arising from anthropogenic $CO_2$ emissions requires a detailed understanding of the carbonate system's natural variability. Coastal ecosystems vary over short spatial and temporal scales, so their dynamics are not well-described by long-term and broad regional averages. A year-long time series of $pCO_2$, temperature, salinity, and currents is used to quantify the high-frequency variability of the carbonate system at the mouth of the Bay of Fundy, Nova Scotia. The seasonal cycle of $pCO_2$ is modulated by a diel cycle that is larger in summer than in winter, and a tidal contribution that is primarily M2, with amplitude roughly half that of the diel cycle throughout the year. The interaction between tidal currents and carbonate system variables leads to lateral transport by tidal pumping, which moves alkalinity and DIC out of the bay, opposite to the mean flow in the region, and constitutes a new feature of how this strongly tidal region connects to the larger Gulf of Maine and Northwest Atlantic carbon system. These results suggest that tidal pumping could substantially modulate the coastal ocean's response to global ocean acidification in any region with large tides and spatial variation in biological activity, requiring that high-frequency variability be accounted for in assessments of carbon budgets of coastal regions.

## 1 Introduction

Oceanic uptake of anthropogenic carbon dioxide ($CO_2$) moderates the rise of atmospheric $CO_2$ concentrations and leads to changes in the ocean carbon cycle. $CO_2$ uptake acidifies the oceans, potentially altering ecosystems and leading to adverse effects for the societies and industries that depend on them. Assessing the vulnerability of these resources to long-term change in ocean acidification requires a detailed understanding of the carbonate system's natural variability that any anthropogenic trend may add to or alter. Coastal ecosystems vary on shorter temporal and spatial scales than the shelf or open ocean so

predicting how they may change under future climate scenarios is more difficult, and requires an understanding of the system's high-frequency components.

The Bay of Fundy (Fig. 1, inset) is an approximately 200 km long, 50 km wide, 75 m deep bay that extends northeastward into Canada from the Gulf of Maine in the Northwest Atlantic Ocean. The regional circulation flows southward along the Scotian Shelf as the Nova Scotia Current, follows the coastline around southern Nova Scotia to enter the southern side of the Bay of Fundy, and exits the bay along the northern coast to join the Eastern Maine Coastal Current in the Gulf of Maine (Bigelow, 1927; Greenberg, 1983; Hannah et al., 2001; Pettigrew et al., 2005; Dever et al., 2016). The mean circulation and water properties of the Gulf of Maine and the Nova Scotia Current are well described (e.g. Smith, 1989; Smith et al., 2001; Hannah et al., 2001; Houghton and Fairbanks, 2001; Aretxabaleta et al., 2008; Gledhill et al., 2015; Dever et al., 2016). Within the bay the mean flow recirculates cyclonically around the outer Bay of Fundy (Aretxabaleta et al., 2008). The geometry of the basin makes it resonant with the M2 tidal frequency and generates the highest tidal range in the world, over 16 m at the head of the bay (Garrett, 1972; Greenberg, 1983). Turbulence from the fast tidal flows keeps much of the basin well-mixed, while the deeper regions of the outer bay develop seasonal stratification. The largest freshwater source to the bay is the St. John River, on the northern coast, so owing to river plume dynamics and the general circulation of the region, this river water primarily propagates along the northern coast, into the Gulf of Maine.

Seasonal and inter-annual variability in air-sea $CO_2$ flux, along with the key carbon cycling processes, have been documented both upstream from the Bay of Fundy on the Scotian Shelf (e.g. Shadwick et al., 2010, 2011; Thomas et al., 2012; Craig et al., 2015), and downstream in the Gulf of Maine (Salisbury et al., 2009; Vandemark et al., 2011). Within the bay, salt marshes have been shown to be weak emitters of $CO_2$ to the atmosphere (Magenheimer et al., 1996), but these results might not be representative of fully submerged regions. Basin-wide estimates of $CO_2$ fluxes have come as part of larger-scale coastal studies where the Bay of Fundy is included as part of the Gulf of Maine region. Using historical $pCO_2$ data and satellite algorithms, Signorini et al. (2013) identified the Gulf of Maine as a weak source of $CO_2$ to the atmosphere. In contrast, results from a coupled biogeochemical-circulation model indicate the Gulf of Maine to be a relatively strong sink of $CO_2$ compared to surrounding areas (Cahill et al., 2016).

The coastal carbon budget and thus the rate at which alkalinity will be able to buffer future ocean acidification depend on the exchange processes between coastal and open oceans. In shallow marshy estuaries, tidal budgets have been estimated for oxygen from dissolved oxygen measurements (Nidzieko et al., 2014) and for carbon from a linear model based on pH and oxidation-reduction potential (Wang et al., 2016). However, to the best of our knowledge, the tidal transport of carbon in this macrotidal system, or far from large freshwater or nutrient sources, has never been addressed.

To investigate this issue, a year-long, high-frequency time series of $pCO_2$, temperature, salinity, and currents was measured via a cabled-to-shore platform in Grand Passage, a tidal channel at the mouth of the Bay of Fundy, Nova Scotia (Fig. 1). This location is ideal for tracing the main input of water into the Bay from the Scotian Shelf. We quantify carbonate system variability on hourly to seasonal timescales, unravel the interaction between the daily and tidal cycles, determine the phase relationship between tidal currents and carbonate system variables, and estimate lateral transports by tidal pumping, which moves alkalinity and dissolved inorganic carbon (DIC) out of the bay, opposite to the mean flow in the region.

## 2 Measurements and data processing

Grand Passage is narrow passage (1 km wide, 4 km long) separating Brier Island from Long Island at the end of the Digby Neck peninsula, which juts out into the mouth of the Bay of Fundy from the southwest corner of Nova Scotia (Fig. 1, inset). The tidal range is 5 to 6 m and peak tidal velocities range from 2 to 3 m s$^{-1}$.

### 2.1 Time series measurements

A year-long record of high-frequency measurements of $p$CO2 (Fig. 2a), temperature (Fig. 2b), and pressure, four months of salinity (Fig. 2c), and one month of velocity data were collected via a cabled-to-shore observatory on a bottom frame that was deployed in approximately 10.5 m mean water depth (location indicated on Fig. 1).

The primary instrument for this experiment was a CONTROS HydroC $CO_2$ sensor, which uses non-dispersive infrared spectrometry (NDIR) to measure gas concentrations that have equilibrated across a hydrophobic membrane. The HydroC was calibrated by the manufacturer before and after deployment and has a resolution of $< 1$ µatm, an accuracy of $\pm 1\%$, and a response time of 65 s at 15° C and 70 s at 5° C. All field measurements fell within the calibrated measurement range of 200 - 1000 µatm. The HydroC was mounted 1 m above the sea floor and cabled to shore for continuous power and data transfer. It recorded $p$CO$_2$ every 1 s from March 2015 to April 2016. The instrument was zeroed every 64 minutes until 16 June 2015, and every 735 minutes for the remainder of the experiment. During zeroing the gas stream is isolated from the membrane and CO$_2$ is removed. Zero-channel values indicate no sensor drift over the deployment period. Following zeroing, partial pressure re-equilibrated over roughly one hour and data from these periods were omitted from analyses.

A CTD was collocated with the HydroC and recorded salinity, temperature, and pressure data every 30 s from March through July 2015. The CTD was recovered and redeployed to record data every 60 s until April 2016. The conductivity sensor bio-fouled soon after redeployment so only temperature and pressure are available for the second half of the experiment.

Salinity measurements were despiked and a linear drift of 0.41 over the first deployment period was removed. The drift was determined by matching the final measured tidal cycle to the initial tidal cycle values from the redeployment of the CTD on the following day (prefouling). Salinity was estimated for the remainder of the year using tidal harmonic analysis (Pawlowicz et al., 2002; Codiga, 2011) of the available four months of data. Salinity variation at frequencies lower or higher than the tidal harmonics is absent from the latter part of the data set, but do not change the final results of this work because the direct effect of salinity and alkalinity (Sect. 2.4) variation is subtracted from the DIC$_{ex}$ variable (Sect. 3.2) used for quantitative analyses.

An RDI Workhorse 600 kHz ADCP was deployed nearby, in approximately 27 m water depth, for part of the experimental period. The upward-looking ADCP recorded velocity in 0.5 m bins at 2 Hz from April 8 - 26, 2015.

All time series variables were low-pass filtered with a cutoff period of 5 min and then subsampled at a 5 min interval.

### 2.2 Bottle samples

Bottle samples were collected from the ferry wharf on the west side of Grand Passage over two days in February 2016. On 16 Feb 2016, 12 bottles were filled hourly from 07:30 to 18:30 local time, using a Niskin off the wharf, from 0.5 - 1m below the

surface. On 17 Feb 2016, 22 samples were collected half-hourly from 07:00 to 18:00. The Niskin was used for seven samples, until it broke. Later samples were collected from the adjacent shore, approximately 5 m from shore in 1.5 m water depth, from 0.5 m below the surface. On 18 March 2017, 15 bottle samples were collected between 09:00 and 13:00 ADT (UTC-3) via Niskin casts off the *Nova Endeavor*, alternating between the center, east, and west sides of Grand Passage across a transect adjacent to the cabled instruments.

Bottles samples for total alkalinity analysis were poisoned with a supersaturated mercuric chloride solution to halt biological activity, and stored for later analysis. Total alkalinity was analyzed by potentiometric titration on a Versatile Instrument for the Determination of Titration Alkalinity (VINDTA 3C). Analytical methods were based on Dickson et al. (2007), including the use of certified reference materials for regular instrument calibration.

## 2.3 Other data sources

Hourly wind speeds and daily precipitation for the entire experiment period were obtained from the Environment and Climate Change Canada (ECCC) weather station on Brier Island, on the western side of Grand Passage (http://climate.weather.gc.ca/climate_data/hourly_data_e.html?StationID=10859).

Atmospheric $CO_2$ concentrations were obtained from the NOAA Greenhouse Gas Marine Boundary Layer Reference data product (Dlugokency et al., 2015). At the time of data processing, ~weekly (1/48 of a year) values were available through 2015 for both zonal average values for 44 - 45° N and global average. Monthly global average values were additionally available through September 2016 (ftp://aftp.cmdl.noaa.gov/products/trends/co2/co2_mm_gl.txt) so estimates for 44 - 45° N were constructed for Jan - Apr 2016 by adding the difference between the zonal and global values for 2013 - 2015, averaged by calendar-week, to the global values for 2016.

Alkalinity and salinity data from the Gulf of Maine shown in Fig. 2f and used in Eq. 1 were obtained from the NOAA Ocean Acidification Program (Salisbury, 2017).

Alkalinity and salinity data from the Scotian Shelf shown in Fig. 2f and used in Eq. 1 were collected at all stations as part of the Atlantic Zone Monitoring Program (AZMP) (Fisheries and Oceans Canada, 2013-2014) in Fall 2013 and Spring 2014. Samples were collected and processed following the methods described in Shadwick and Thomas (2014) for the 2007 AZMP data set.

## 2.4 Estimating alkalinity from salinity

A linear relationship between salinity, S, and alkalinity, TA, (Fig. 2f) was calculated with data from the bottle samples collected at the field site in February 2016 (n = 34) and March 2017 (n = 15), from the AZMP Fall 2013 (n = 357) and Spring 2014 (n = 467) cruises, and from the Gulf of Maine in 2015 (n = 286).

$$TA = (45.75 \pm 0.39)S + (709.72 \pm 12.76) \tag{1}$$

The Grand Passage data covers a small (<1 ΔPSU) salinity range so a regression from only the Grand Passage field site was not robust. The Grand Passage salinity-alkalinity relationship was not a priori expected to be identical to the data collected 10

to 200 km offshore in the Scotian Shelf and Gulf of Maine regions, but Fig. 2f shows there is no significant change in the water mass end members between those regions, which are up- and downstream of Grand Passage in the regional circulation pattern. A time series of alkalinity (Fig. 2c) is calculated from salinity using Eq. 1.

## 2.5    Calculating DIC

5    DIC concentration and pH (Fig. 2d,e) for the carbonate system at equilibrium were calculated from measurements of pCO2, salinity, alkalinity, and temperature, with constants following Dickson and Millero (1987) and Weiss (1974). We used van Heuven's (2011) MATLAB code for Lewis and Wallace's (1998) 'CO2SYS' implementation of the equations for carbonate equilibria.

## 3    Results and Discussion

10    **3.1    Seasonal evolution of carbonate system variables**

Grand Passage connects two adjacent embayments, St. Mary's Bay and the Bay of Fundy, and because the water properties in these embayments can differ, the strong semi-diurnal (M2) tide causes tidal period variability in the carbonate system. $pCO_2$ varies on annual, daily, and tidal time scales, and the size of the daily and tidal signals also changes with the season. The seasonal evolution of $pCO_2$ (Fig. 2a) is dominated by the effect of temperature (Fig. 2b), rising in summer and declining in 15    winter, while biological processes only modulate this cycle. $pCO_2$ ranges from a minimum of 307 µatm in spring 2015 up to a maximum of 557 µatm in early fall, with daily and M2 period variation of 32 µatm and 10 µatm, respectively. In November, the daily range drops to 11 µatm and $pCO_2$ decreases throughout the winter. Final measured values in March 2016 were 30 µatm higher than the those measured the previous spring, owing to the difference in water temperature.

Temperature (Fig. 2b) has a strong seasonal cycle, increasing from 2 °C in April 2015 up to 14 °C in September. Temperature 20    decreases from October 2015 through March 2016, when it reaches an annual minimum of 3.5 °C, 1.5 degrees higher than the previous springtime minimum. The ranges of daily and M2 tidal variation are 0.35 °C and 0.7 °C, respectively, in the summer, and 0.1 °C and 0.15 °C during fall through spring.

The salinity (Fig. 2c, right axis) average for our March - July period of data availability was 31.9 with a 0.18 tidal variation and no daily signal. Salinity was not correlated with local wind or precipitation. Alkalinity mean and tidal variation were 2177 25    and 8.24 $\mu mol\,kg^{-1}$, respectively, based on the linear relationship between salinity and alkalinity measured with bottle samples (Fig. 2f and Sect. 2.4).

The in situ $pCO_2$ value is important because it determines the air-sea flux of $CO_2$, but is not an ideal variable to assess biogeochemical carbonate dynamics because of its dependence on temperature and alkalinity, which obfuscate the biological processes. DIC (Fig. 2d) does not have a strong temperature dependence so it better depicts the biological DIC variation. DIC 30    declines steeply during the spring bloom, and then increases though the winter. During the spring bloom and throughout the summer there is a large daily range in DIC. In October, daily variability shrinks, and DIC increases steadily throughout the

winter. pH varies as an approximate inverse of $p\text{CO}_2$. pH has an average value of 8.01 ($= -\log_{10} < [H^+] >$) and daily and M2 variation are equivalent to changes in pH of 8 to 8.03 and 8 to 8.01, respectively.

## 3.2 Unraveling daily and tidal cycles of biogeochemically-driven changes in DIC

In order to isolate the biogeochemically mediated changes in DIC from the variability due to purely physical processes, we
define the variable 'excess DIC' as the DIC remaining after the concentration expected from the physical mixing of water masses of different salinities and alkalinities has been subtracted. The DIC dependence on salinity and alkalinity, $\text{DIC}_{\text{mix}}$, is estimated numerically with CO2SYS using fixed values of $p\text{CO}_2$ and temperature. $\text{DIC}_{\text{mix}}$ includes both a salinity-dependent component and a background constant that depends on $p\text{CO}_2$ and temperature. The mean $p\text{CO}_2$, 446 µatm, and temperature, 8 °C, for the deployment period are chosen for the calculation. $\text{DIC}_{\text{ex}}$ is calculated by subtracting $\text{DIC}_{\text{mix}}$ from the full (observed)
DIC, $\text{DIC}_{\text{obs}}$ (Fig. 3).

$$\text{DIC}_{\text{ex}} = \text{DIC}_{\text{obs}} - \text{DIC}_{\text{mix}} \qquad \text{where} \qquad \text{DIC}_{\text{mix}} = \text{DIC}(p\text{CO}_2 = 446, \text{T} = 8, S, \text{TA}) \tag{2}$$

Only the time-variation of the resulting $\text{DIC}_{\text{ex}}$ is meaningful, not the absolute value. This time-variation is not sensitive to the choice of fixed $p\text{CO}_2$ and temperature from within the ranges typical of this field site. The relationship between $p\text{CO}_2$ and DIC that is captured in $\text{DIC}_{\text{ex}}$ is unchanged over the small natural range of alkalinity in this region, so $\text{DIC}_{\text{ex}}$ is not
sensitive to the choice of a measured, tidal harmonic, or even constant salinity estimate. $\text{DIC}_{\text{ex}}$ is presumed to be predominantly biogeochemically-driven, but also includes any changes in DIC due to air-sea exchange, which we could not calculate on daily time scales, but are shown to be small on weekly time scales in section 3.5.

$\text{DIC}_{\text{ex}}$ (Fig. 4a) decreases rapidly April and May due to the spring bloom, and more gradually over the summer, as explained by Craig et al. (2015). Following a decline in October (yearday 270-290) suggestive of a fall bloom, $\text{DIC}_{\text{ex}}$ increases steadily
through fall and winter. The daily cycle of $\text{DIC}_{\text{ex}}$ is evident throughout year when highlighted by use of color to indicating local time of day in Fig. 4a. Daily maxima occur in early morning ($\sim$06:00, pink dots) following nighttime respiration, and minima in late afternoon ($\sim$17:00, green dots) following the peak hours of sunlight and photosynthesis. This cycle is shifted several hours earlier than the one reported on the Scotian Shelf by Thomas et al. (2012). The daily range of $\text{DIC}_{\text{ex}}$ is approximately three times as large in summer as in winter, consistent with higher summer sunlight supporting higher phytoplankton growth.
The solar and tidal cycles combine to create a fortnightly cycle (visible in Fig. 4a, but more clear in Fig. 4b). This beating pattern in the time series is due to the difference between the 12.42 h M2 tidal and 24 h diel frequencies. Mathematically, this effect is identical to a spring-neap tidal variation, but here the daily cycle is due to solar insolation, rather than the solar gravitational force. The diagonal banding pattern in Fig. 4b shows the M2 tide progressing 50 minutes later each day. The daily morning peak and afternoon low appear as broad horizontal stripes, and are most visible for yeardays 100-275. A similar daily
pattern has been observed on the Scotian Shelf (Thomas et al., 2012), but no tidal signal was detected there. The pulsing of the strength of the morning high and afternoon low is due to the coincidence of an M2 maximum or minimum with the time of the daily minimum or maximum.

The overlapping daily and tidal cycles apparent in Fig. 4 can be separated and quantified by spectral analysis. Power spectral densities of $DIC_{ex}$, $P_{DIC_{ex}}(f)$ for frequency $f$, were calculated by Welch's method on detrended time series for each month with eight approximately week-long Hamming windowed segments with 50% overlap. The position of peaks in spectra of month-long $DIC_{ex}$ time series (Fig. 5a) identify the frequencies with the greatest variability, and the area under each peak equals the contribution of that frequency to the total signal variance. These February and August examples show a large daily peak and a slightly smaller M2 tidal peak for both months, and show that $DIC_{ex}$ is more variable in August than February at all frequencies. The third and fourth peaks visible in Fig. 5a are harmonics of the 24h and M2 frequencies and do not substantially contribute to the total signal variance. The variance of $DIC_{ex}$ at the 24 h and M2 frequencies are calculated from the area under the spectra using a 5-point peak width, $\sigma^2_{DIC_{ex}} = \sum P_{DIC_{ex}}(f)\Delta f$. The total $DIC_{ex}$ variance is ten times higher in August than in February but in both seasons, the daily and M2 frequencies represent the majority of the variability: 56% (56%) and 19% (7%), respectively, of the total $DIC_{ex}$ signal variance in August (February).

The strengths of M2 and daily cycles in $DIC_{ex}$ evolve with the season (Fig. 5b). The range of $DIC_{ex}$ variation at a particular frequency is defined as $2A$ for amplitude $A$, i.e. the "peak to trough" difference of a sine wave. The RMS values of a sinusoid equals $A/\sqrt{2}$, so the range of the daily or tidal DIC cycle equals $2\sqrt{2}\sqrt{\sigma^2_{DIC_{ex}}}$. Months from April through September have a similar size daily signal near 15 µmol kg$^{-1}$, with lower values throughout the winter. The tidal variation is always smaller than the daily variation, but is also generally larger in summer and smaller in winter. However, unlike the daily signal, June and July have smaller tidal variation than the shoulder months of April, May, August and September. Changes in the strength of tidal signal reflects changes in spatial gradients of $DIC_{ex}$, presumably owing to season fluctuations in the spatial variation of biological activity.

## 3.3 Tidal phasing

The relationship between the tidal flow and $DIC_{ex}$, temperature, salinity/alkalinity, and $H^+_{ex}$ are depicted in Fig. 6 for April 2015, when velocity measurements were available. This tidal phase information complements the daily cycle of $DIC_{ex}$ emphasized in Fig. 4a.

Salinity and corresponding alkalinity (Fig. 6a) are lowest during late flood, and highest between max ebb and early flood (see ebb/flood directions on Fig. 1). Salinity values are lower during flood compared to those on ebb, which indicates that the water from St. Mary's Bay/Scotian Shelf that enters the Bay of Fundy through Grand Passage each flood tide is fresher than what exits during ebb. This tidal asymmetry in salinity holds for all months of the year.

Temperature (Fig. 6b) is lowest in late flood and peaks a short time later during early ebb, and overall the water is colder during flood than during ebb. Unlike the salinity asymmetry, the temperature asymmetry changes sign with the seasons. The shallower St. Mary's Bay is more sensitive to surface heat fluxes than the deeper Bay of Fundy, so it is warmer in spring and summer and colder in fall and winter. As a result, the oscillating tides move heat into the Bay of Fundy half of the year, and out half the year.

$DIC_{ex}$ (Fig. 6c) peaks at low slack and the lowest values occur during late flood and early ebb. $H^+_{ex}$ (Fig. 6d) also peaks at low slack and has the lowest values during early ebb. This pattern suggests lower net community production in the bay than on

the shelf, likely a result of a stronger spring bloom on the shelf than in the bay, which is advected by the mean currents around southern Nova Scotia. Smaller scale spatial variation in nutrient or light availability owing to different water depths or mixing rates could also contribute to different growth rates on the two sides of Grand Passage.

## 3.4 Lateral transport by tidal pumping

Transport of carbon through Grand Passage is driven both by net volume transport and by *tidal pumping* - oscillatory tides moving water masses with different properties back and forth on each tidal cycle. Water volume flux per meter of channel width, $q$ ($\mathrm{m^2 s^{-1}}$) $= \overline{u}h$, for water depth $h$ (m) and depth-averaged velocity $\overline{u}$ ($\mathrm{m\ s^{-1}}$) can be decomposed into a time-mean ($<>$) and fluctuating ($'$) part, $q = <q> + q'$. By definition, there is no net water volume transport by the time-varying volume flux used to calculate tidal pumping. The fluctuations in volume transport (Fig. 7a) that drive tidal pumping vary in magnitude over the spring-neap cycle, but return to zero each tidal cycle.

Any correlation between the tidal water volume flux, $q'$, and fluctuations in conserved carbonate system concentration variables, $S' = S - <S>$ ($\mathrm{g\ m^{-3}}$ or $\mathrm{mol\ m^{-3}}$), leads to a scalar flux by tidal pumping, $Q_{pump}^S = <q'S'>$ ($\mathrm{mol\ m^{-1}s^{-1}}$ or $\mathrm{g}$ $\mathrm{m^{-1}s^{-1}}$), when averaged over times scales longer than a tidal cycle. Salinity is nearly vertically uniform at this field site (Razaz et al., 2018), and we assume DIC is also vertically uniform because the vertical mixing time scale is much shorter than the time scale of gas exchange owing to the high turbulence in the Bay of Fundy (appendix A1), i.e. $\int_0^h S(z)u(z)dz = S\overline{u}$. Fluctuations of the conserved variable that are not correlated with fluctuations in water volume flux, such as the daily or seasonal cycles shown in Fig. 4a, are in $S'$ but do not contribute to $Q_{pump}^S$ because they are not correlated with $q'$.

Cumulative along-channel transports, $M$ ($\mathrm{m^3}$, mol, or g), of water volume, alkalinity, and $\mathrm{DIC_{ex}}$ (Fig. 7) are calculated by multiplying $q'$, $q'\mathrm{TA}'$, and $q'DIC'_{ex}$ by the width ($w$ = 800 m) of Grand Passage, and integrating in time, $t$. We assume spatial uniformity of water properties across the section.

$$M_S(t) = \int_0^t q'S'w\,dt \tag{3}$$

The fluctuating water volume flux, $q'$, used to compute these transports is a tidal harmonic solution based on a fit to the month of ADCP data from April 2015. The harmonic fit represents 93% of the observed variability (i.e. $R = 0.97$) in water volume flux. By contrast, the tidal harmonic fit to the four months of salinity measurements (Sect. 2.1) is not well-correlated with the observations ($R = 0.35$) owing to the non-sinusoidal shape of the salinity signal as well as longer period variability. Due to this poor fit, alkalinity fluxes are not computed for the period without direct salinity measurements. $\mathrm{DIC_{ex}}$ is not affected by the salinity, as described in Sect. 3.2, so $\mathrm{DIC_{ex}}$ transport is calculated for the full year.

For the first half of 2015, salt and alkalinity (Fig. 7b) are pumped southward, out of the Bay of Fundy by the tides. $\mathrm{DIC_{ex}}$ (Fig. 7c) also has net negative (southward) transport over this period, but has shorter periods of near zero or positive transport in March, mid-April, and early June.

Tidal pumping for $\mathrm{DIC_{ex}}$ is also calculated for the second half of the deployment year (Fig. 7d) and continues the negative trend until January 2016, when the flux becomes slightly positive for the last two months of the measurement period. Salinity

from a seasonal climatology (e.g Richaud et al., 2016; Signorini et al., 2013) suggests that the sign of the lateral salinity gradient is the same all year, indicating that the sign of the alkalinity flux will stay the same throughout the year.

Notably, these transports move alkalinity and $DIC_{ex}$ in the opposite direction of the mean flow of the region, which follows the coast clockwise around southwestern Nova Scotia, moving northward into the Bay of Fundy near Grand Passage (e.g. Aretxabaleta et al., 2008). Within Grand Passage, salinity/alkalinity transport by tidal pumping is roughly 20% of that by the mean volume flux through the channel (appendix A2). The exact fraction may vary outside the channel, but the important point is that tidal pumping likely plays a first order role in carbon transport budgets anywhere with large tides and spatial gradients in the biogeochemical water properties, including the entire width of the mouth of the Bay of Fundy.

While the mean volume transport through Grand Passage cannot be extrapolated to the full width of the Bay of Fundy, the tidal pumping term could plausibly apply over the eastern side of the mouth, where salinity gradients are positive into the bay. If the transport by tidal pumping is applied out to just 10 km from shore ($\sim$15% of the width of the mouth of the bay), which is a $\sim$50 m deep region, the $DIC_{ex}$ transported out of the Bay of Fundy through lateral advection by tidal pumping would be $5\times10^8$ kg in a year. This value is three times what is estimated to leave the Bay of Fundy by outgassing to the atmosphere if the Grand Passage air-sea fluxes (Sect. 3.5) applied over the whole bay.

## 3.5  Air-sea CO$_2$ flux

High-frequency variability in $DIC_{ex}$ is assumed to be driven by biological and biogeochemical processes, but air-sea flux plays a significant role on long time scales. We assess the importance of air-sea flux to the carbon budget by calculating weekly and annual fluxes, and the equivalent changes in DIC. Oceanic $pCO_2$ was lower than atmospheric $pCO_2$ (Fig. 2a) for the first two months of observations, April and May, and then rose and remained higher than atmospheric $pCO_2$ for the following ten months, June through March, giving an annual net negative $CO_2$ flux (i.e. outgassing) at this site.

Atmospheric and oceanic $CO_2$ concentrations (Fig. 2a) and wind speed (Sect. 2.3), are used to calculate the flux of $CO_2$ between the atmosphere and ocean at the field site. Atmospheric and oceanic $pCO_2$ data were interpolated onto the hourly wind time base. Air-sea flux, $F$ (mol m$^{-2}$ s$^{-1}$), is calculated following Wanninkhof et al. (2009) Eq. 3, here using the convention that $F$ is positive for a flux of gas from the atmosphere into the ocean.

$$F = -kK_0(pCO_{2w} - pCO_{2a}) \tag{4}$$

where $K_0$ is the solubility of $CO_2$ (mol m$^{-3}$ Pa$^{-1}$) and $k$ (m s$^{-1}$) is gas transfer velocity. $k$ (cm h$^{-1}$) can be represented well for wind speeds, $U$ (m s$^{-1}$), below 15 m s$^{-1}$ at Schmidt number $Sc = 660$ with the empirical formula (Wanninkhof et al., 2009, Eq. 37)

$$k_{660} = 0.24\langle U_{10}^2\rangle \tag{5}$$

$U_{10}$ is the 10 m wind speed measured at the ECCC station on Brier Island. U10 was also computed as the difference between the air and water speeds (up to $\pm 2$ m s$^{-1}$), which increased the air-sea flux estimates by approximately 5%. Air-sea flux is initially computed on an hourly time scale to fully capture the quadratic wind-speed dependence, but only weekly (or longer) averages of $F$ are robust owing to the weekly time scale of the available atmospheric pCO2 data.

The average air-sea $CO_2$ flux at this site is $-4.6 \times 10^{-8}$ mol m$^{-2}$ s$^{-1}$, which is similar in magnitude to previous Gulf of Maine flux estimates, but with outgassing most of the year rather than the more even split between positive and negative fluxes reported at a deeper site (Vandemark et al., 2011) or for regional averages (Signorini et al., 2013; Cahill et al., 2016). If this gas exchange was spatially uniform over the Bay of Fundy (approx. $10^4$ km$^2$), this air-sea flux would release $1.75 \times 10^8$ kg carbon (=$1.45 \times 10^{10}$ mol) into the atmosphere per year.

The annual local flux is equivalent to a $-47$ µmol kg$^{-1}$ change in DIC$_{ex}$ (Eq. 6) when applied to 30 m water depth in Grand Passage over the year-long deployment.

$$\Delta \text{DIC}_{\text{air-sea}} = \frac{F_{\text{air-sea}}}{h} \Delta t \tag{6}$$

The weekly averaged flux is typically between 0 and $-1 \times 10^{-7}$ mol m$^{-2}$ s$^{-1}$, which is equivalent to up to $-2$ µmol kg$^{-1}$ change in DIC$_{ex}$ over a week. The maximum weekly value occurred in late September 2015, and was $-2 \times 10^{-7}$ mol m$^{-2}$ s$^{-1}$, yielding a $-4$ µmol kg$^{-1}$ change in DIC$_{ex}$ in one week.

### 3.6 Consideration of the local DIC budget

Observed DIC$_{ex}$ variation is due to the difference between local (water column and sediment) net community production (NCP), local air-sea flux, and advective changes due to spatial gradients of biological production or air-sea flux (Appendix A3). Daily variation in DIC$_{ex}$ can reasonably be attributed to the local time rate of change, and the tidal variation is likely advective. Longer cycles of variation could have both a local seasonal cycle and advective contributions from seasonal variation occurring weeks or months upstream.

We did not directly measure spatial gradients of DIC$_{ex}$, but can infer that the local along-channel gradient is positive (increasing northwards) from the observation that the highest and lowest values tend to occur near low and high slack-water, respectively (Fig. 6). A positive along-channel DIC$_{ex}$ gradient carried by a northward regional circulation decreases local DIC$_{ex}$.

DIC$_{ex}$ returns near to its initial value over the full annual cycle (Fig. 4a) so the annual mean $\partial \text{DIC}_{ex}/\partial t$ is zero. The outgassing air-sea flux and along-channel advection both decrease DIC$_{ex}$, so to close the DIC budget (Eq. A6) the biologically-driven change in DIC must be positive (community NCP < 0) on average over the year, even though it is negative in the spring.

### 4 Conclusions

Open ocean carbon cycles can be described by annual and daily variations, and taken to be uniform over large spatial scales, but near the coast, ecosystems change markedly over short distances, so the dynamics of the carbonate system cannot be well-described without resolving the spatial variation or capturing the effects of lateral advection.

We unravel tidal advection from the diel production and respiration cycles due to solar radiation, and show that the tides pump both alkalinity and DIC out of the Bay of Fundy into the open ocean, opposite to the transport via the residual flow. This

tidal pumping process, in contrast to any residual flow, does not result in a net transport of water, but can greatly enhance the exchange of passive scalars like DIC. In regions with strong tidal currents and spatial variation in biological activities, tidal pumping could substantially modulate the coastal ocean's response to global ocean acidification.

High-frequency measurements reveal previously unaccounted for tidal variations in the carbonate system, in addition to more precise quantification of the familiar daily cycles and annual blooms. These results are crucial prerequisites to a deeper understanding of coastal systems' resilience or vulnerability to anthropogenic change.

*Data availability.* The data that support the findings of this study are available from the corresponding author upon reasonable request.

## Appendix A

### A1   Vertical mixing

In this highly turbulent, unstratified water column, the vertical mixing time scale (Eq. A1) is much shorter than the time scale of gas exchange (Eq. A2).

$$T_{\mathrm{mix}} = \frac{h^2}{A_\nu} \sim \frac{h^2}{(\kappa u_* h/12)} = \frac{h}{\kappa u_*/8} \approx \frac{25\,\mathrm{m}}{0.4 \times 0.1\,\mathrm{m/s} \times 0.125} = 5000\,\mathrm{s} = 1.4\,\mathrm{h} \tag{A1}$$

$$T_{\mathrm{air\text{-}sea}} = \frac{h}{k_{660}} \approx \frac{25\,\mathrm{m}}{4 \times 10^{-5}\,\mathrm{m/s}} = 6.25 \times 10^5\,\mathrm{s} = 7\,\mathrm{days} \tag{A2}$$

where $h$ is total water depth at the ADCP site, $k_{660}$ value is average from the dataset, $A_\nu$ is eddy viscosity estimated as the mid water depth value of a cubic eddy viscosity profile for a logarithmic near-bed velocity profile and zero surface wind stress (e.g. Lentz, 1995), $\kappa$ is the von Karmen constant, $u_*$ is shear velocity with value chosen from McMillan et al.'s (2013) Grand Passage site "GP2" very close to the one used in the present study.

### A2   Mean volume transport

The mean volume flux, $< q >$, is derived from both the mean depth-averaged along-channel velocity and the mean volume transport due to asymmetries in the tidal cycle $< q > = < \overline{u}h > = < \overline{u} > < h > + < \overline{u}'h' >$. At the ADCP location, $< \overline{u} > = 0.036\,\mathrm{m\,s^{-1}}$ and $< h > = 26.6\,\mathrm{m}$, which are indicated by the solid black lines in Fig. 6, yielding $0.95\,\mathrm{m^2\,s^{-1}}$ volume flux.

The position of the data points in Fig. 6 compared to the mean axes made by $< \overline{u} >$ and $< h >$ values, shows that the water tends to be deeper during ebb than during flood. If the tide was a perfect standing wave (slack at high and low water, and max ebb and flood speeds occurring at the mean water depth), the shape of the depth-velocity points would form a circle. The mean volume flux due to this tidal asymmetry, $< \overline{u}'h' > = -1.61\,\mathrm{m^2\,s^{-1}}$, which is larger than, and in the opposite direction of, the volume flux by the mean depth-averaged velocity.

Assuming spatial uniformity over the channel width, the volume flux driven by the mean depth-averaged velocity applied to the mean salinity of 31.9 yields a salt flux of $2.3 \times 10^4 \text{ kg s}^{-1}$, and the volume flux driven by tidal asymmetry generates a salt flux of $-4.0 \times 10^4 \text{ kg s}^{-1}$.

The salt flux driven by total $< q >$ from both components is $-1.6 \times 10^4 \text{ kg s}^{-1}$, which is five times the salt flux by tidal pumping shown in Fig. 7. However, the mean velocity and especially the phase lag between the tide depth and velocity is generated by friction and therefor sensitive to the specific channel geometry (e.g. Geyer and MacCready, 2014) and may vary across the mouth of the Bay of Fundy.

## A3 Budget for biological DIC production

The change in observed $\text{DIC}_{\text{ex}}$, is given by

$$\frac{\partial \text{DIC}_{\text{ex}}}{\partial t} = \frac{\partial \text{DIC}_{\text{bio}}}{\partial t} + \frac{\partial \text{DIC}_{\text{air-sea}}}{\partial t} \tag{A3}$$

Biology and air-sea exchange both drive local changes in time, and both could potentially have changes in time due to advection of spatial gradients, which is what we (almost definitely) observe over the tidal cycle. e.g.

$$\frac{\partial \text{DIC}_{\text{bio}}}{\partial t} = \frac{\partial \text{DIC}_{\text{bio,local}}}{\partial t} - u \frac{\partial \text{DIC}_{\text{bio}}}{\partial x} \tag{A4}$$

where $x$ is along-channel distance, and $\partial \text{DIC}_{\text{bio,local}}/\partial t$ corresponds to local carbon consumption (i.e. equal to $-$NCP). So in total

$$\frac{\partial \text{DIC}_{\text{ex}}}{\partial t} = \frac{\partial \text{DIC}_{\text{bio,local}}}{\partial t} + \frac{\partial \text{DIC}_{\text{air-sea,local}}}{\partial t} - u \frac{\partial \text{DIC}_{\text{bio}}}{\partial x} - u \frac{\partial \text{DIC}_{\text{air-sea}}}{\partial x} \tag{A5}$$

If we assume the air-sea flux is spatially uniform, $u \frac{\partial \text{DIC}_{\text{air-sea}}}{\partial x} = 0$, then the change in observed DIC that is attributable to biological activity, $\text{DIC}_{\text{bio}}$, is given by

$$\frac{\partial \text{DIC}_{\text{bio,local}}}{\partial t} - u \frac{\partial \text{DIC}_{\text{bio}}}{\partial x} = \frac{\partial \text{DIC}_{\text{ex}}}{\partial t} - \frac{\partial \text{DIC}_{\text{air-sea}}}{\partial t} \tag{A6}$$

*Author contributions.* R.M.H. performed the research and wrote the manuscript with the guidance of H.T., who, with A.E.H., initially conceived of the project. R.M.H. and H.T. developed analysis trajectory. A.E.H. provided velocity, temperature, salinity, and pressure data. R.C. and W.B. programmed, deployed, and maintained the instruments. J.S. provided Gulf of Maine salinity and alkalinity data. All authors consulted on the manuscript.

*Competing interests.* The authors have no competing interests.

*Acknowledgements.* This work was funded by the MEOPAR Ocean Acidification program and the Environment and Climate Change Canada (ECCC) Gulf of Maine Initiative. Thanks to Walter Judge for instrument preparation and deployment, to Brittany Curtis and Jonathan Lemay for water sample collection, and Mike Huntley and his crew on the *Nova Endeavor*.

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

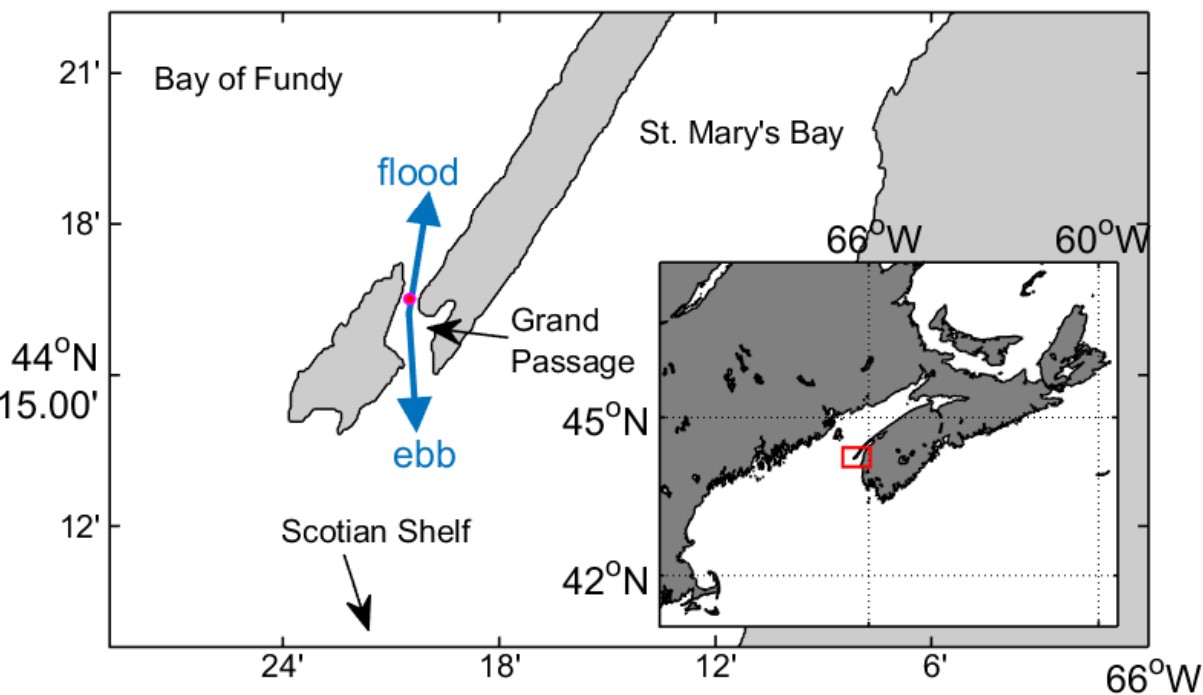

**Figure 1.** Map of Grand Passage, which cuts through a peninsula in southwest Nova Scotia. Instrument location indicated by pink dot. Flood and ebb tide flow directions shown blue arrows. Inset: Map of the region with red box indicating field site at the mouth of the Bay of Fundy.

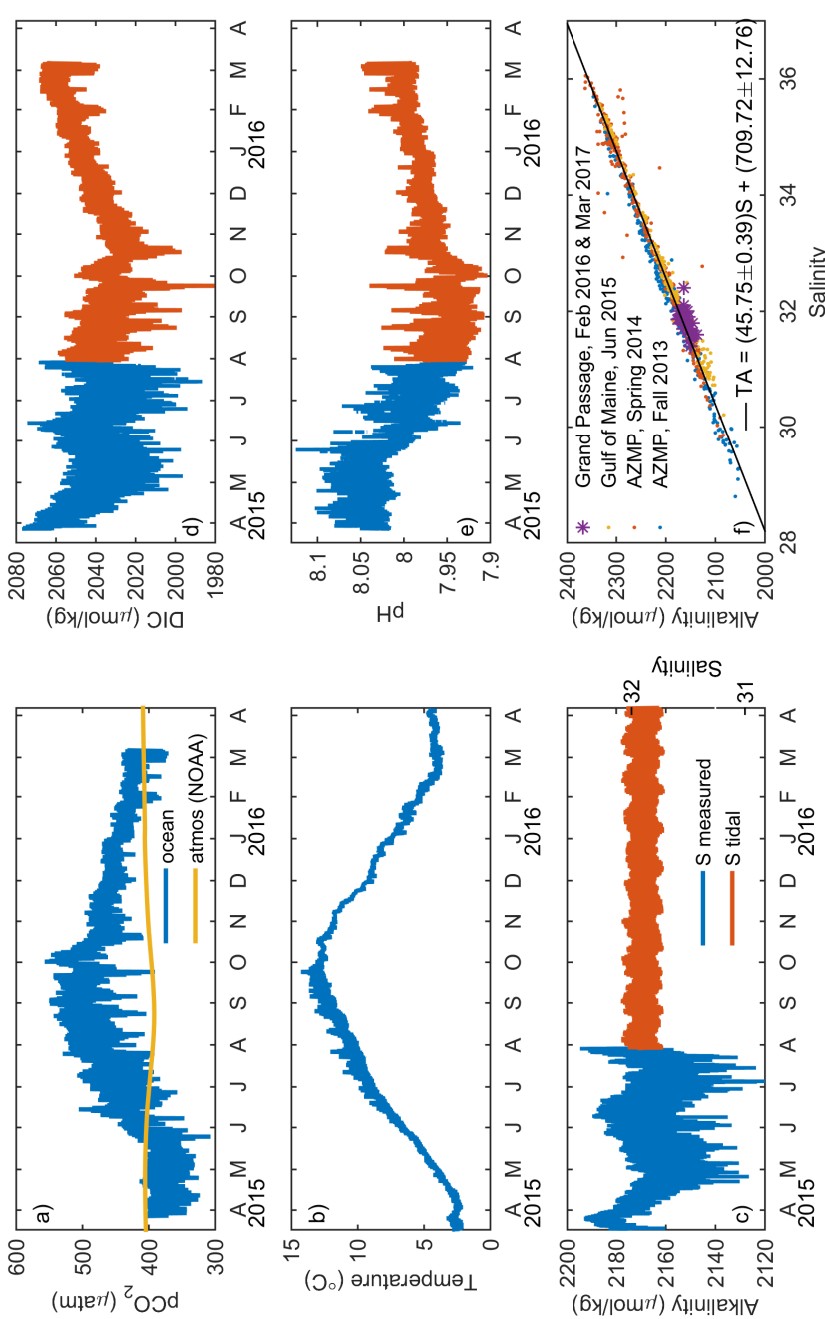

**Figure 2.** (a–c) Measured variables and (d,e) timeseries generated from the carbonate system equilibrium solution for data shown in (a–c). (a) $pCO_2$ in Grand Passage (blue) and NOAA's weekly atmospheric zonal average for 44–45° N (yellow) (b) Temperature (c) Salinity (right y-axis) and alkalinity (left y-axis) from periods with salinity data (blue) and generated from a tidal prediction when measurements were not available owing to instrument failure (orange) (d) DIC (e) pH. (f) Alkalinity vs. salinity from bottle samples at the Grand Passage field site in 2016 and 2017 (purple) as well as from the Scotian Shelf via the Atlantic Zone Monitoring Program in 2013 (blue) and 2014 (orange), and the Gulf of Maine in 2015 (yellow). Linear regression (black) used to generate alkalinity shown in (c)

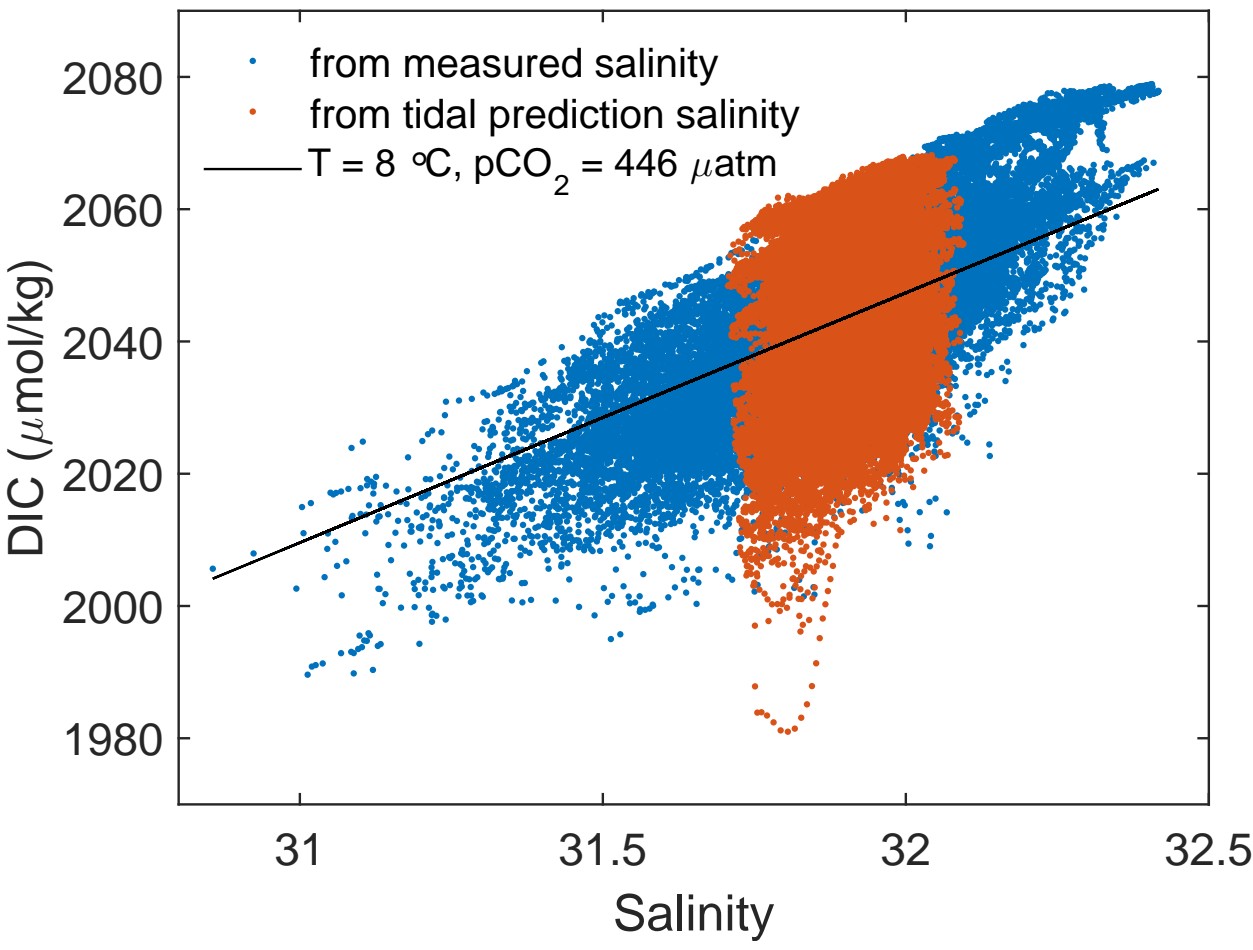

**Figure 3.** DIC vs. salinity for measured (blue) and tidal reconstruction (orange) carbonate system solutions. DIC at fixed $p\mathrm{CO}_2$ (446 $\mu atm$) and temperature (8°C) (black) is subtracted from observed DIC to create $\mathrm{DIC}_{ex}$.

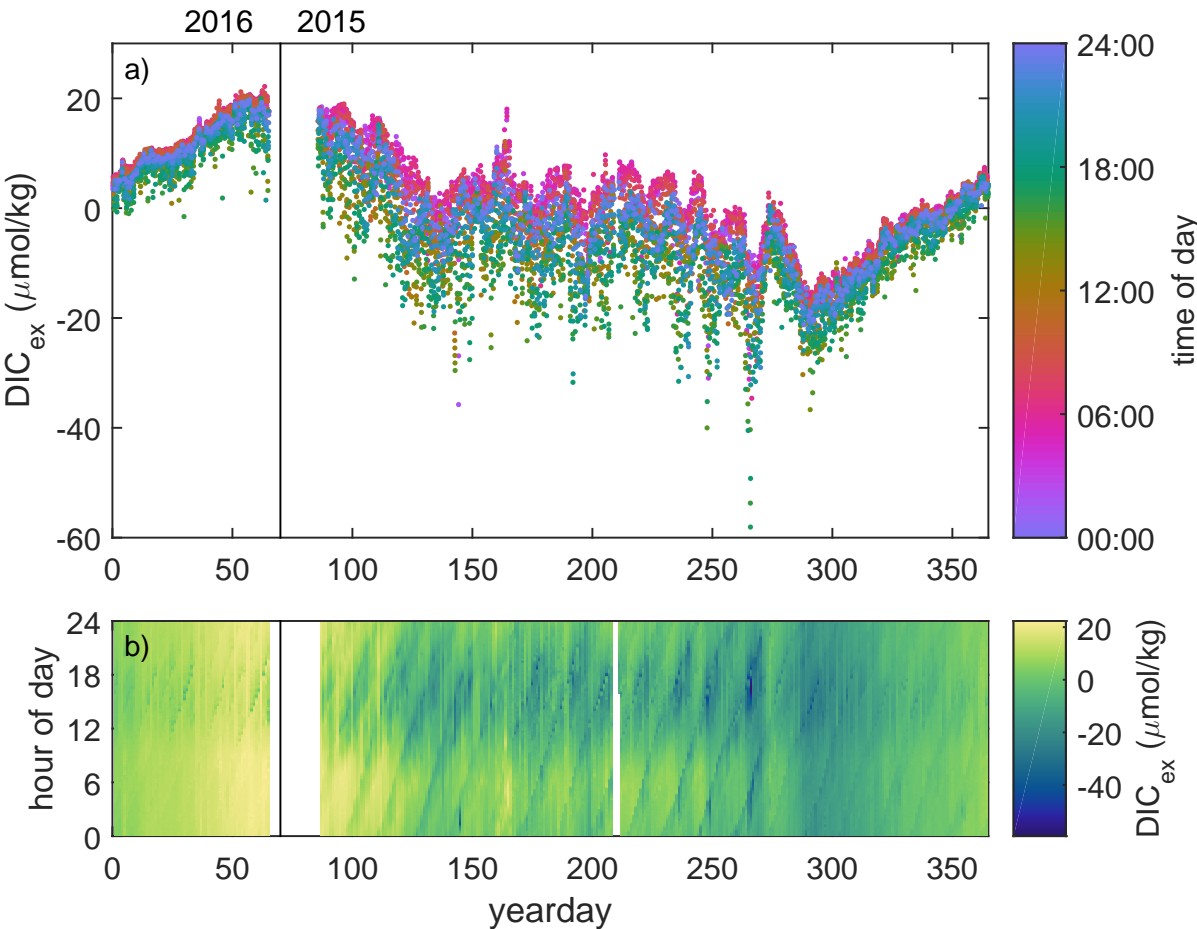

**Figure 4.** (a) DIC$_{ex}$ for a calendar year with local time of day indicated by color. The time-variation of the resulting DIC$_{ex}$ is meaningful, while the total value depends on the specific choice of fixed $p$CO$_2$ used for the computation. (b) DIC$_{ex}$ (color) for each hour of the day over a calendar year.

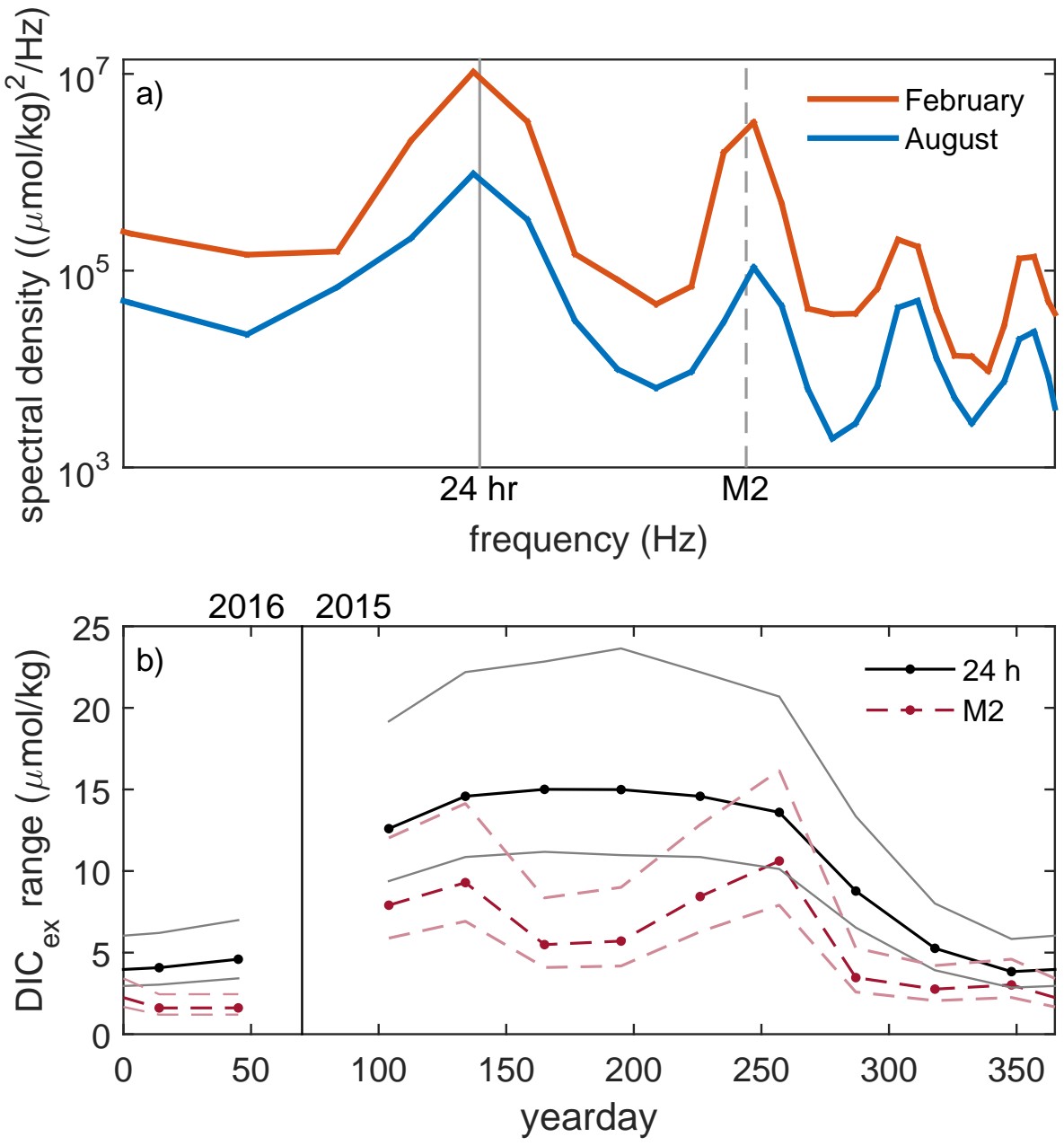

**Figure 5.** (a) Spectra of $DIC_{ex}$ in February (blue) and August (orange). (b) $DIC_{ex}$ range over daily (black, solid) and M2 (red, dashed) cycles for each month of the year. Amplitude determined from area beneath peaks in spectra, such as examples shown in (a). 95% confidence intervals in lighter colors.

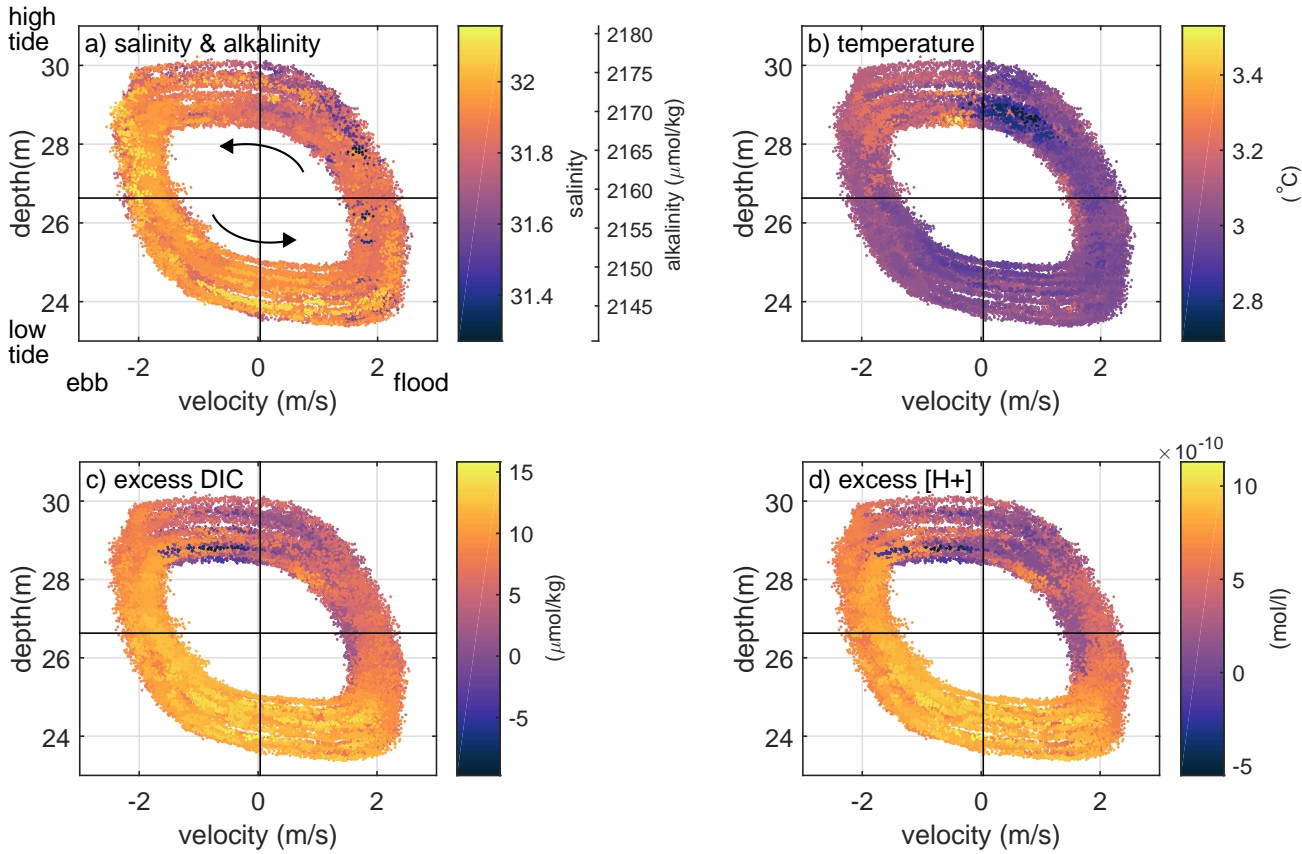

**Figure 6.** The relationship between the tidal flow and (a) salinity/alkalinity, (b) temperature, (c) DIC$_{ex}$, and (d) H$^+_{ex}$ are depicted for April 2015, when velocity measurements were recorded. The phase of the tide is indicated by ADCP water depth on the y-axis and depth-average along-channel velocity on the x-axis (positive in flood direction, i.e. northward, into the Bay of Fundy), with mean values indicated by black lines. Phase progression is counter-clockwise, shown by arrows in (a). Scalar concentration is indicated by color, e.g. more yellow data points visible on the left side of (a) indicates higher salinity during ebb tide, when water is leaving the Bay of Fundy. The the variation in position of the tidal ellipse between each pass of the tidal cycle reflects the changes in tidal range and maximum speed associated with the spring-neap cycle. For all variables shown by colors, 18-hr high pass filtered values plus mean are plotted to visually highlight the M2 variability by eliminating the daily signal; the filtering is not used for data analyses. H$^+_{ex}$ is computed by the same method as DIC$_{ex}$, for H$^+ = 10^{-pH}$.

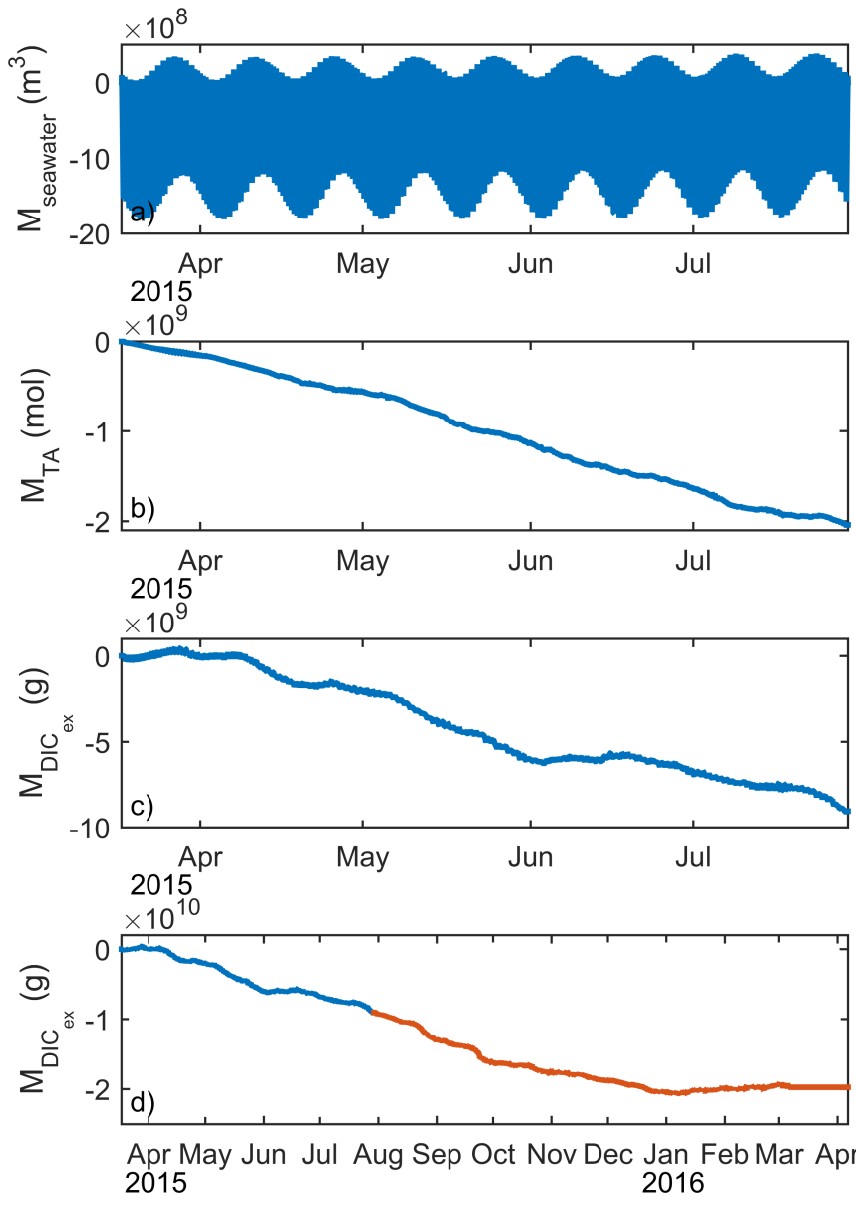

**Figure 7.** Cumulative along-channel transport since the start of the deployment period by tidal pumping of (a) water volume, (b) alkalinity, and (c) $DIC_{ex}$ for March-July 2015 and (d) $DIC_{ex}$ for the full year; orange color indicates period not shown in (a-c). Salt or alkalinity fluxes are not computed for the full year because fluxes from the constructed tidal salinities will only reflect the mean trend already seen in Fig. 7b and not contain changes in phase or magnitude that might occur in the fall or winter.