# Peer review of "High-frequency variability of CO2 in Grand Passage, Bay of Fundy, Nova Scotia"

_Biogeosciences, 2018_

## Referee Comment (RC1) · Anonymous Referee #1 · 12 Sep 2018

1. General Comments: Tides are often viewed as symmetric with respect to volume flow, so the effects of tidal pumping are misunderstood or ignored. Given the discrepancy of biogeochemical estimates to determine whether this region is a net source or sink of $CO_2$ to the atmosphere, this study gives essential insight to the contribution of high-frequency variability to the carbon cycle, specifically carbon transport via tidal pumping. However, I am not yet convinced that air-sea flux has been given full consideration. The relationship between $DIC_{ex}$ and air-sea flux is still unclear to me even after reading through the appendix. With respect to data presentation, I recommend the abandonment of the rainbow colormap and the adoption of color schemes more true to the data and accessible to colorblind readers.

2. Specific Comments:

- P3, l15: The tidally-estimated salinity has a much lower range than the measured salinity. How sensitive are your downstream calculations, especially $DIC_{ex}$, to this reduction in variability in salinity and hence alkalinity?

- P3, l23 - 25: Alkalinity samples were taken only during the daytime. I presume they spanned the entire tidal range?

- P4, l28 - 29: While the full equation is not necessary, it would be helpful to enumerate what measurements were used to calculate DIC and pH.

- P5, l6 - 7 and Fig 2a, b: Generally, I would expect any gas to decrease in the summer and increase in the winter due to solubility. Is there something special about $CO_2$ that I am missing here?

- P6, l6: Air-sea exchange is very difficult to calculate, especially in a dynamic system such as this one. How is it treated in CO2SYS? Is it the Wanninkhoff parameterization shown in the discussion? Should the reader refer to the appendix at this point?

- P6 l14 - 21: There seems to be abrupt changes in the nature of the $DIC_{ex}$ data around days 265 and 290. Any explanations?

- P6 l26 - 28 and Fig 5: In addition to the 24-hr and M2, there are two peaks. From the x-scale, it is difficult to tell whether they are important or merely harmonics of the predominant frequencies.

- P7 l19: By "lower growth," do you mean lower primary production or photosynthesis? Growth can also be heterotrophic.

- P7 l24 - P8 l10: Provide units of the variables given to aid dimensional analysis.

- P9 l27 - 29: Did you finish your thought here? Do you have an estimate of advection? There's no period.

- P12 l7: Given the spatial variability in $pCO_2$, S, and T, can you assume spatially-uniform air-sea flux?

- Fig 3: There is a tail of low DIC in the tidally-predicted salinity points. Are these anomalous?

- Fig 4: I appreciate how differently a) and b) portray the exact same data. I would recommend not using a rainbow colormap for 4a. It is good that it wraps around, but the data are not represented linearly (green spans 6 hours) and it is not accessible for readers who are colorblind. Take a look at other cyclic color maps. I understand why you might want to put the winter 2016 data at the front of the figure, but there is nothing significant about January 1st with respect to solar or lunar cycles. It's better to arrange the data chronologically and smoothly.

- Fig 6: These plots are extremely informative, but again, rainbow colormaps are ineffective at communicating data. For a) and b), use the parula colormap used in Fig 4b. For c) and d), try a diverging map (blue - white - red, for example).

3. Technical Corrections:

- P2, l1: Transpose "of" and "understanding."
- P2, l20: Remove the redundant "of."
- P3, l5 - 11: Descriptions of sampling frequency alternate between s and Hz. Choose one for consistency.
- P4, l3 - 4: Double check this link. I got a 404 error.
- P4, l18 - 19: Generally the past tense is preferred here. Feel free to use the first person.
- P5, l17: Eliminate the space in "relation ship."

__

- P5, 19 - 20: These two sentences make it sound as if air-sea flux was a biological process.
- P5, l30: What software is used to run CO2SYS?
- P7, l9 - 12: Use past tense.
- Fig 6a: Should axis should read "Salinity" rather than "salt?"

---

## Referee Comment (RC2) · Anonymous Referee #2 · 5 Oct 2018

Review of Horwitz et al., High-frequency variability of CO2 in Grand Passage, Bay of Fundy, Nova Scotia.

This manuscript presents an interesting analysis of the complex physical and biogeochemical interactions that control the marine CO2 system in a region with extreme tidal currents. It deserves publication even if the relevance for global change issues such as ocean acidification is low, but this should not be the criterion for the scientific quality of a paper. Still, I suggest some revisions which can improve the readability, intelligibility and scientific correctness of the manuscript.

Overall structure: I suggest to include a short paragraph that characterizes the hydrography of the Bay of Fundy. Readers outside of North America may not be familiar with the hydrographic biogeochemical conditions in that region. The "Discussion" contains

two rather short sections (4.2 and 4.3). My suggestion is to merge these with "Results" which is then named "Results and Discussion". The CO2 flux estimate (4.1) is of no relevance for the science in your manuscript. To satisfy those who are always hunting for such numbers, move it to the Appendix.

Comments: 2.1 Time series measurements: The pCO2 measurements are the basis of your study. There-fore it is necessary to give some more information about the measuring principle, preci-sion/accuracy, response time, calibration procedure, etc. . What is meant with "flushing"?

2.4 Estimating alkalinity from salinity, p.4/line 23 " Grand passage measurements are not expected . . .", unclear sentence, one can only guess what the meaning is. Has the intercept any biogeochemical meaning?

2.5 Calculating DIC, p.4/last line: The "system of equations" is not created by Lewis and Wallace (1998), but represents the well-known thermodynamics of acid-base equilibria. Ko from Weiss (1974)?

3.1 Seasonal evolution of measured and equilibrium solution variables "equilibrium so-lution variables", bad term in the headline.

p. 5/line 17: Tidal alkalinity variation of 718 $\mu$mol/kg??? Check. p.5/last paragraph: How are the first two sentences logically related to each other? DIC is a conservative variable and does not depend on temperature. The observed DIC decrease refers to different water masses, it is therefore not allowed to make any conclusions about effect of "photosynthesis and respiration". Such changes occur on a background DIC level which is directly related to the alkalinity. In order to identify any biogeochemical DIC changes, it is thus necessary to remove the effect of the differing back-ground DIC as you have done it by calculating DICex.

3.2 Unravelling daily and tidal cycles of biogeochemical driven changes in DIC Sub-script "mix" is somewhat misleading, I prefer to call it "background DIC". Please make

clear that the zero level in Fig. 4 has no biogeochemical meaning since it de-pends on the choice of the reference pCO2. You could have taken also the atmospheric pCO2 and spring SST as reference because in many ocean regions the spring bloom starts when the surface water is approximately at equilibrium with the atmosphere. In that case the zero DICex means "zero " biology. p.6/2nd paragraph: Your interpretation of the seasonal DICex pattern is confined to biologi-cal effects and ignores the effect of gas exchange. Can you roughly estimate the relative im-portance of the two pro-cesses?

3.3 Tidal phasing The calculation of H+ex makes no sense because H+ is a non-conservative variable and does follow conservation of mass. This is also reflected in a strong dependency of $\Delta H+/\Delta TA$ and thus of H+ex on the choice of the reference pCO2.

---

## Author Comment (AC1) · 7 Nov 2018

The comment was uploaded in the form of a supplement:
https://www.biogeosciences-discuss.net/bg-2018-378/bg-2018-378-AC1-supplement.pdf

---

## Author Response (AR1)

Thank you to both Referees for providing specific and constructive comments. They have helped improve this manuscript. Below, we respond to each comment from **Referee #1**, **Referee #2**, and the **Associate Editor**, in that order. Reviewer comments are in blue. Quoted manuscript text is in *italics*, with new additions in *green*. A marked-up manuscript that shows all changes made since the original submission follows our response.

**Referee #1**

1. General Comments:
I am not yet convinced that air-sea flux has been given full consideration. The relationship between DICex and air-sea flux is still unclear to me even after reading through the appendix.
We have addressed specific comments below, but in general the relationship between air-sea flux and DICex is shown in section 3.5 (formerly section 4.1). We do not have our own atmospheric pCO2 data, so are limited to weekly or longer temporal scales for the air-sea flux. We (now) show the airsea flux is small on weekly time scales. It is almost always negative, so accumulates to a large effect on annual time scales. Because we do not have the spatial data to fully close a DIC budget, we can only infer that over the annual cycle, local biology plus advective effects must balance the air-sea flux.

With respect to data presentation, I recommend the abandonment of the rainbow colormap and the adoption of color schemes more true to the data and accessible to colorblind readers.
Colormaps have been updated to perceptually uniform ones that are colorblind/greyscale friendly

2. Specific Comments:
• P3, l15: The tidally-estimated salinity has a much lower range than the measured salinity. How sensitive are your downstream calculations, especially DICex, to this reduction in variability in salinity and hence alkalinity?

The salinity range has negligible effect on DICex because the DIC dependence on pCO2 is effectively unchanged over our observed salinity range. In fact, we get nearly identical results if we just use a reasonable constant salinity. Below, the DIC dependence on pCO2 for a fixed salinity is shown for S = 31 (blue) and S = 32 (green). There is an offset between the lines – this DIC dependence on salinity is exactly what is subtracted when we create the DICex variable – but the curvature is nearly identical. The orange line below is the S = 32 line shifted downward by a constant. The slight difference between blue and orange lines is the size of the error that remains if we use an estimated salinity value that is ~1 PSU different than the true value we were unable to measure. The CTD failure would have been much more problematic if we worked in an estuary with a large enough salinity range so substantially change the DIC-pCO2 relationship. Alkalinity *does* depends strongly on salinity (unlike DICex), so we do not estimate alkalinity fluxes in Fig 7 for the period without salinity measurements.

[Figure]

• P3, l23 - 25: Alkalinity samples were taken only during the daytime. I presume they spanned the entire tidal range?

Yes, by sampling for 12 hours each day, the bottle samples covered a full tidal cycle on each of two sampling days, including high and low water times, and max ebb and flood.

• P4, l28 - 29: While the full equation is not necessary, it would be helpful to enumerate what measurements were used to calculate DIC and pH.

We have updated the text to specify measurements used for the calculation:

*"DIC concentration and pH (Fig. 2d,e) for the carbonate system at equilibrium were calculated from measurements of pCO2, salinity, alkalinity, and temperature, with constants following Dickson and Millero (1987) and Weiss (1974). We used van Heuven's (2011) MATLAB code for Lewis and Wallace's (1998) 'CO2SYS' implementation of the equations for carbonate equilibria."*

• P5, l6 - 7 and Fig 2a, b: Generally, I would expect any gas to decrease in the summer and increase in the winter due to solubility. Is there something special about CO2 that I am missing here?

pCO2 increases with increasing temperature because Henry's Law constant changes with temperature. Partial pressure is the atmospheric (headspace) CO2 gas pressure with which the dissolved CO2(aq) would be in equilibrium. Gas pressure increases with increasing temperature, so for a given dissolved CO2 molar concentration, a higher equilibrium gas pressure (ie partial pressure) is required at a higher temperature. Takahashi et al., 2002 (https://doi.org/10.1016/S0967-0645(02)00003-6) is a good example of pCO2 T-dependence. On the other hand, at a given pCO2, the resulting equilibrium concentration of DIC will be higher at cold temperatures than at warm ones.

There is nothing special about CO2 except perhaps that partial pressure is the common way to report it, while other gases like oxygen are more often reported as a concentration.

• P6, l6: Air-sea exchange is very difficult to calculate, especially in a dynamic system such as this one. How is it treated in CO2SYS? Is it the Wanninkhoff parameterization shown in the discussion? Should the reader refer to the appendix at this point?

Air-sea flux is not a part of the CO2SYS carbonate system equations. CO2SYS calculates the thermodynamic equilibrium speciation of the seawater carbonate system. CO2SYS does not compute transport processes.

Yes, the Wanninkhof parameterization is cited and reproduced as our equations 4 & 5 in section 3.5 (formerly numbered section 4.1). In section 3.2, we have added a reference to Section 3.5, where the calculation of ∆DIC resulting from air-sea flux is shown.

*"DICex is presumed to be predominantly biogeochemically driven, but also includes any changes in DIC due to air-sea exchange, which we could not calculate on short time scales, but are shown to be small on daily time scales in section 3.5."*

We calculate the fluxes hourly to capture the quadratic relationship between wind speed and gas flux, but must average the hourly fluxes on weekly (or longer) time scales for them to be accurate because that is the time-resolution of the NOAA atmospheric pCO2 data product. The Wanninkhof flux formula depends linearly on ∆pCO2, so the weekly average of our estimate should be correct. Because of this data limitation, we do not attempt to assess air-sea flux at daily or tidal time scales. We have updated section 3.5 (formerly 4.1) to include the typical weekly ∆DIC change owing to air-sea flux to emphasize its small effect on DIC on short time scales, and clarify our justification of attributing high frequency changes in DIC to biological processes.

*"The weekly averaged flux is typically between 0 and -1e-7 mol/m^2/s which is equivalent to up to -2 umol/kg change in DICex change over a week. The maximum weekly value occurred in late September 2015, and was -2e-7 mol/m^2/s yielding a -4 umol/kg change in DICex in one week."*

• P6 l14 - 21: There seems to be abrupt changes in the nature of the DICex data around days 265 and 290. Any explanations?

We don't know for sure, but there are several factors that may have contributed. The low value at day 265 doesn't appear to be related to any weather event and appears to follow the fortnightly beating pattern. The steady decrease in DICex from days 273-290, and subsequent rise may be related to a regional fall bloom. Our site is never stratified, but a bloom could have been triggered on the nearby shelf by a moderate wind event on day 273. This is also the time of year when water temperature transitions from rising to declining and reduced temperature variability (Fig 2b). Unfortunately the region was often cloudy so satellite-based chlorophyll estimates do not show these rapid transitions.

• P6 l26 - 28 and Fig 5: In addition to the 24-hr and M2, there are two peaks. From the x-scale, it is difficult to tell whether they are important or merely harmonics of the predominant frequencies.

Yes, these additional 2 peaks are harmonics of the 24h and M2 frequencies. They are an order of magnitude smaller than the primary peaks so do not substantially contribute to the total signal variance. We have noted this in the text:

*"These February and August examples show a large daily peak and a slightly smaller M2 tidal peak for both months, and show that DICex is more variable in August than February at all frequencies. The third and fourth peaks visible in Fig. 5a are harmonics of the 24h and M2 frequencies and do not substantially contribute to the total signal variance. The variance of DICex at the 24 h and M2 frequencies are calculated from the area under the spectra…"*

• P7 l19: By "lower growth," do you mean lower primary production or photosynthesis? Growth can also be heterotrophic.

DIC only reflects net community productivity and so we cannot distinguish lower photosynthesis versus higher respiration. We have updated the sentence to use "lower net community production."

• P7 l24 - P8 l10: Provide units of the variables given to aid dimensional analysis.
We have added units each time a variable is introduced in section 3.4: q ($m^2s^{-1}$), u (m s$^{-1}$), h (m), S (mol m$^{-3}$ or g m$^{-3}$), Q (mol m$^{-1}$s$^{-1}$ or g m$^{-1}$s$^{-1}$), M (m$^3$, mol, or g)

• P9 l27 - 29: Did you finish your thought here? Do you have an estimate of advection? There's no period.

Thank you, the period was missing after DICex at the end of line 29. We cannot quantitatively estimate advection's contribution to the local DIC budget because we do not have data on the along-channel gradient of DIC. We only feel comfortable inferring the sign of the gradient from the single-point time series data and the purpose is to constrain our discussion of the DIC budget. This very short section (4.2) has been merged with section 4.3 (DIC budget discussion) and the merged section is now numbered 3.6. The paragraphs are rearranged, but the text is unchanged.

• P12 l7: Given the spatial variability in pCO2, S, and T, can you assume spatially-uniform air-sea flux?

Over some small area, yes. We assume yes for the purpose of simplifying the equation, and then can decide how broadly we can apply our simplified system. We think this is a reasonable assumption within Grand Passage, where the water is well mixed vertically and laterally. We would love to be able to assess these parameter across the entire Bay of Fundy, but more spatial data is needed and, for this reason, we only include back-of-the-envelope type budget approximations in the discussion section of the paper.

• Fig 3: There is a tail of low DIC in the tidally-predicted salinity points. Are these anomalous?

This is the event on Day 265 that you asked about earlier. As far as we know it's real, and is a result of a real, low pCO2 measurement on that day. It's the only time mid-year that the measured pCO2 value dips below the atmospheric value (blue spike below yellow line in late September, Fig 2a).

• Fig 4: I appreciate how differently a) and b) portray the exact same data. I would recommend not using a rainbow colormap for 4a. It is good that it wraps around, but the data are not represented linearly (green spans 6 hours) and it is not accessible for readers who are colorblind. Take a look at other cyclic color maps. I understand why you might want to put the winter 2016 data at the front of the figure, but

there is nothing significant about January 1st with respect to solar or lunar cycles. It's better to arrange the data chronologically and smoothly.

Fig 4a has been changed to a perceptually uniform, cyclic colormap based on the "phase" colormap (*cmocean* package, Thyng et al, Oceanography, 2016). Fig 4b has also been changed to the perceptually uniform "haline" *cmocean* colormap. Both colormaps are accessible to those with moderate red-green colorblindness and haline works in greyscale. We understand that a Jan 1 start to the year is not specifically related to tidal, daily, or annual biogeochemical cycles, but we choose that start because it is the conventional way of viewing a year and also highlights what fraction of a full annual cycle we missed with our deployment/recovery dates, and the close match of the start and end DICex values even though the pCO2 was quite different on account of warmer water that spring (shown & explained w/ Fig 2).

[Figure]

• Fig 6: These plots are extremely informative, but again, rainbow colormaps are ineffective at communicating data. For a) and b), use the parula colormap used in Fig 4b. For c) and d), try a diverging map (blue - white - red, for example).

The colormap has been switched to the perceptually uniform "thermal" *cmocean* colormap. We do not use a diverging colormap for (c) or (d) because the zero value is not meaningful. We use the same

colormap for all four subfigures to aid the reader in viewing the differences in how the four water properties vary around the tidal cycle.

[Figure]

3. Technical Corrections:
• P2, l1: Transpose "of" and "understanding." Fixed.

• P2, l20: Remove the redundant "of." Fixed.

• P3, l5 - 11: Descriptions of sampling frequency alternate between s and Hz. Choose one for consistency.
We switched sampling *"at 1 Hz"* to *"every 1 s"*, and also at the end of section 2.1, switched *"frequency of 0.003 Hz"* to *"period of 5 min"*

• P4, l3 - 4: Double check this link. I got a 404 error.
This link works for me without an error. However, it did point to the main page for weather station, which required an extra click to get to the hourly data. The link has been updated to point directly to the hourly data type we used, and I will check with the editor to make sure a functional link is published. http://climate.weather.gc.ca/climate_data/hourly_data_e.html?StationID=10859

• P4, l18 - 19: Generally the past tense is preferred here. Feel free to use the first person. Fixed.

• P5, l17: Eliminate the space in "relation ship." Fixed.

• P5, 19 - 20: These two sentences make it sound as if air-sea flux was a biological process.
We have rewritten these sentences as
*"The in situ pCO2 value is important because it determines the air-sea flux of CO2, but is not an ideal variable to assess biogeochemical carbonate dynamics because of its dependence on temperature and alkalinity, which obfuscate the biological processes."*

• P5, l30: What software is used to run CO2SYS?
We used the MATLAB version. This info has been added next to the citation for the code in the Calculating DIC methods section 2.5:
*"DIC concentration and pH (Fig. 2d,e) for the carbonate system at equilibrium were calculated from measurements of pCO2, salinity, alkalinity, and temperature, with constants following Dickson and Millero (1987) and Weiss (1974). We used van Heuven's (2011) MATLAB code for Lewis and Wallace's (1998) 'CO2SYS' implementation of the equations for carbonate equilibria."*

• P7, l9 - 12: Use past tense. While the measurements occurred in the past, we presume the findings hold presently.

• Fig 6a: Should axis should read "Salinity" rather than "salt?" Fixed. Also 'alk' is now written fully as alkalinity

**Referee #2**

Overall structure:
I suggest to include a short paragraph that characterizes the hydrography of the Bay of Fundy. Readers outside of North America may not be familiar with the hydrographic biogeochemical conditions in that region.

We have added a new paragraph to the introduction:

*"The Bay of Fundy (Fig. 1, inset) is an approximately 200 km long, 50 km wide, 75 m deep bay that extends northeastward into Canada from the Gulf of Maine in the Northwest Atlantic Ocean. The regional circulation flows southward along the Scotian Shelf as the Nova Scotia Current, follows the coastline around southern Nova Scotia to enter the southern side of the Bay of Fundy, and exits the bay along the northern coast to join the Eastern Maine Coastal Current in the Gulf of Maine (Bigelow, 1927; Greenberg, 1983; Hannah et al., 2001; Pettigrew et al., 2005; Dever et al., 2016).* The mean circulation and water properties of the Gulf of Maine and the Nova Scotia Current are well described (e.g. Smith, 1989; Smith et al., 2001; Hannah et al., 2001; Houghton and Fairbanks, 2001; Aretxabaleta et al., 2008; Gledhill et al., 2015; Dever et al., 2016). *Within the bay the mean flow recirculates cyclonically around the outer Bay of Fundy (Aretxabaleta et al., 2008). The geometry of the basin makes it resonant with the $M2$ tidal frequency and generates the highest tidal range in the world, over 16 m at the head of the bay (Garrett, 1972; Greenberg, 1983). Turbulence from the fast tidal flows keeps much of the basin well-mixed, while the deeper regions of the outer bay develop seasonal stratification. The largest freshwater source to the bay is the St. John River, on the northern coast, so owing to river plume dynamics and the general circulation of the region, this river water primarily propagates along the northern coast, into the Gulf of Maine."*

The "Discussion" contains two rather short sections (4.2 and 4.3). My suggestion is to merge these with "Results" which is then named "Results and Discussion".

We have followed your suggestion. The former section 4.1 (air-sea flux) is now sections 3.5. We have combined the two very short sections (4.2, 4.3) into a single one (3.6), now titled "*Consideration of the local DIC budget*" to emphasize that this section is a qualitative discussion rather than quantitative result.

The CO2 flux estimate (4.1) is of no relevance for the science in your manuscript. To satisfy those who are always hunting for such numbers, move it to the Appendix.

We understand your point that the air-sea flux is not directly needed for our result about tidal cycles. However, air-sea flux is needed to justify that most DICex variation is due to biology, and leads towards a better understanding of the local carbon budget, even though we cannot entirely close the budget with the existing data set. Therefore, we have chosen to keep it in the body of the manuscript.

We have added two sentences to the beginning of section 3.5 (formerly 4.1) to explain the purpose of the section:

*"High-frequency variability in DICex is assumed to be driven by biological and biogeochemical processes, but air-sea flux plays a significant role on long time scales. We assess the importance of air-sea flux to the carbon budget by calculating weekly and annual fluxes, and the equivalent changes in DIC."*

Comments:
2.1 Time series measurements: The pCO2 measurements are the basis of your study. Therefore it is necessary to give some more information about the measuring principle, precision/accuracy, response time, calibration procedure, etc. .What is meant with "flushing"?

We have updated section 2.1 to include more information about the instrument and the zeroing and re-equilibration period. We removed the term "flushing"

*"The primary instrument for this experiment was a CONTROS HydroC CO2 sensor, which uses non-dispersive infrared spectrometry (NDIR) to measure gas concentrations that have equilibrated across a hydrophobic membrane. The HydroC was calibrated by the manufacturer before and after deployment and has a resolution of <1 µatm, an accuracy of ±1%, and a response time of 65 s at 15 °C and 70 s at 5 °C. All field measurements fell within the calibrated measurement range of 200 - 1000 µatm. The HydroC was mounted 1 m above the sea floor and cabled to shore for continuous power and data transfer. It recorded pCO2 every 1 s from March 2015 to April 2016. The instrument was zeroed every 64 minutes until 16 June 2015, and every 735 minutes for the remainder of the experiment. During zeroing the gas stream is isolated from the membrane and CO2 is removed. Zero-channel values indicate no sensor drift over the deployment period. Following zeroing, partial pressure re-equilibrated over roughly one hour and data from these periods were omitted from analyses."*

2.4 Estimating alkalinity from salinity, p.4/line 23 "Grand passage measurements are not expected . . .", unclear sentence, one can only guess what the meaning is. Has the intercept any biogeochemical meaning?

This sentence has been updated for clarity
*"The Grand Passage salinity-alkalinity relationship was not a priori expected to be identical to the data collected 10 to 200 km offshore in the Scotian Shelf and Gulf of Maine regions, but Fig. 2f shows there is no significant change in the water mass end members between those regions, which are up- and downstream of Grand Passage in the regional circulation pattern."*
The y-intercept of the linear fit, 710 µmol/kg, represents the alkalinity of the (average) freshwater endmember.

2.5 Calculating DIC, p.4/last line: The "system of equations" is not created by Lewis and Wallace (1998), but represents the well-known thermodynamics of acid-base equilibria. Ko from Weiss (1974)?

The text has been update following your suggestion. Yes, Weiss (1974) is used by *van Heuven's (2011)*.
*"DIC concentration and pH (Fig. 2d,e) for the carbonate system at equilibrium were calculated from measurements of pCO2, salinity, alkalinity, and temperature, with constants following Dickson and Millero (1987) and Weiss (1974). We used van Heuven's (2011) MATLAB code for Lewis and Wallace's (1998) 'CO2SYS' implementation of the equations for carbonate equilibria."*

3.1 Seasonal evolution of measured and equilibrium solution variables "equilibrium solution variables", bad term in the headline.

Section title changed to "Seasonal evolution of carbonate system variables"

p. 5/line 17: Tidal alkalinity variation of 718 µmol/kg???

Thank you very much for catching this typo. The correct value of 8.24 µmol/kg has been added to the revised manuscript.

p.5/last paragraph: How are the first two sentences logically related to each other? DIC is a conservative variable and does not depend on temperature. The observed DIC decrease refers to different water masses, it is therefore not allowed to make any conclusions about effect of "photosynthesis and respiration". Such changes occur on a background DIC level which is directly related to the alkalinity. In order to identify any biogeochemical DIC changes, it is thus necessary to remove the effect of the differing back-ground DIC as you have done it by calculating DICex.

We have reworded the first 2 sentences to make a better transition from discussion pCO2 to DIC. We got a little ahead of ourselves, knowing what the DICex results showed later and have deleted the text.

*"The in situ pCO2 value is important because it determines the air-sea flux of CO2, but is not an ideal variable to assess biogeochemical carbonate dynamics because of its dependence on temperature and alkalinity, which obfuscate the biological processes. DIC (Fig. 2d) does not have a strong temperature dependence so it better depicts the biological DIC variation. DIC declines steeply during the spring bloom, and then increases though the winter.* *. During the spring bloom and throughout the summer there is a large daily range in DIC. In October, daily variability shrinks, and DIC increases steadily throughout the winter. pH varies as an approximate inverse of pCO2. pH has an average value of 8.01 (= -log10 < [H+] >) and daily and M2 variation are equivalent to changes in pH of 8 to 8.03 and 8 to 8.01, respectively."*

3.2 Unravelling daily and tidal cycles of biogeochemical driven changes in DIC
Subscript "mix" is somewhat misleading, I prefer to call it "background DIC".

We understand the point – that we call "mix" includes both a constant background value that is not attributed to any particular process, as well as the component correlated to salinity. We think it would be confusing to call it "background" when it varies in time owing to a known process, so we prefer "mix". We considered further dividing the "mix" term into a constant background value plus a zero-mean salinity-dependent variable, but decided adding an extra term would unnecessarily complicate this simple equation.

We have updated the text to further clarify that our DICex includes a background value in addition to the linear salinity dependence:
*"The DIC dependence on salinity and alkalinity, DICmix, is estimated numerically with CO2SYS using fixed values of pCO2 and temperature. DICmix includes both a salinity-dependent component and a background constant that depends on pCO2 and temperature. The mean pCO2, 446 µatm, and temperature, 8°C, for the deployment period are chosen for the calculation. DICex is calculated by subtracting DICmix from the full (observed) DIC, DICobs (Fig. 3)."*

Please make clear that the zero level in Fig. 4 has no biogeochemical meaning since it depends on the choice of the reference pCO2. You could have taken also the atmospheric pCO2 and spring SST as reference because in many ocean regions the spring bloom starts when the surface water is approximately at equilibrium with the atmosphere. In that case the zero DICex means "zero" biology.

We have rearranged the paragraph following Eq 2 to lead with this fact (former first sentence was moved above Eq 2) and emphasize that the mean DICex value is not meaningful, rather than just as a justification for our results not being sensitive to our choice of pCO2, T.

"*Only the time-variation of the resulting DICex is meaningful, not the absolute* value. This *time-variation is not sensitive to the choice of fixed pCO2 and temperature from within the ranges typical of this field site. The relationship between pCO2 and DIC that is captured in DICex is unchanged over the small natural range of alkalinity in this region…*"

p.6/2nd paragraph: Your interpretation of the seasonal DICex pattern is confined to biological effects and ignores the effect of gas exchange. Can you roughly estimate the relative importance of the two processes?

We have estimated the air-sea flux and the equivalent change in DIC in section 3.5 (formerly section 4.1). In the original submission, we only included the annual value. We have now added the weekly change in DIC owing to air-sea flux as well. We cannot estimate the flux and resulting ΔDIC accurately on shorter time scales because the atmospheric pCO2 data available from NOAA is only weekly.

In section 3.2 we have added
"*DICex is presumed to be predominantly biogeochemically driven, but also includes any changes in DIC due to air-sea exchange, which we could not calculate on daily time scales, but are shown to be small on weekly time scales in section 3.5.*"

And in section 3.5 we have added
"*The weekly averaged flux is typically between 0 and -1e-7 mol/m^2/s which is equivalent to up to -2 umol/kg change in DICex change over a week. The maximum weekly value occurred in late September 2015, and was -2e-7 mol/m^2/s yielding a -4 umol/kg change in DICex in one week.*"

Your question specifically mentioned the seasonal cycle, though this section (3.2) is focused on the daily and tidal time scales. The air-sea flux is small on the short time scales, but makes a cumulative change in DICex of -47 umol/kg over the entire year (given in section 3.5). Because the observed DICex returns to near its initial value after a full year, we infer that the biology makes the opposite contribution, putting 47 umol/kg of DIC into the water over the year. However, from this data set, we cannot determine if the long term trend in DICbio is due to advection or local biology, as discussed in section 3.6 (formerly 4.3)

3.3 Tidal phasing The calculation of H+ex makes no sense because H+ is a nonconservative variable and does follow conservation of mass. This is also reflected in a strong dependency of ΔH+/ΔTA and thus of H+ex on the choice of the reference pCO2.

We use [H+]ex where we would have shown pH variation around a tidal period in Fig 6d because it does not make sense to do frequency-base filtering on the time series of a logarithmic variable, nor does it make sense to covert the tidal variation in [H+]ex back into a pH range because the range would depend on the mean.

While protons (mass) are conserved, can be transported, and could be balanced at a constant temperature, we understand that forecasting changes in proton concentration from a proton transport would depend on changes/differences in buffering capacity.

**Associate Editor**

During your revisions, I will have to see more information about maintenance and validation of the pCO2 sensor during the deployment of the present study since these are the foundation of your paper.

We have followed the internationally recommended procedure of having manufacturer calibration performed before and after deployment (Pereira et al., 2017, full citation below). Calibration by the user is not possible for this instrument. As recommended by the manufacturer, the zero-channel values were used to check for sensor drift and we found no significant drift over the deployment period (drift substantially less than the 1% measurement accuracy for seawater pCO2 values).

Text in bold below has been added since the Author's Response to Reviewers to further clarify the instrument performance expectations. We have revised the text in section 2.1:

*"The primary instrument for this experiment was a CONTROS HydroC CO2 sensor, which uses non-dispersive infrared spectrometry (NDIR) to measure gas concentrations that have equilibrated across a hydrophobic membrane. The HydroC was calibrated by the manufacturer* **before and after** *deployment and has a resolution of <1 μatm, an accuracy of ±1%, and a response time of 65 s at 15 °C and 70 s at 5 °C.* **All field measurements fell within the calibrated measurement range of 200 - 1000 μatm.** *The HydroC was mounted 1 m above the sea floor and cabled to shore for continuous power and data transfer. It recorded pCO2 every 1 s from March 2015 to April 2016. The instrument was zeroed every 64 minutes until 16 June 2015, and every 735 minutes for the remainder of the experiment. During zeroing the gas stream is isolated from the membrane and CO2 is removed.* **Zero-channel values indicate no sensor drift over the deployment period.** *Following zeroing, partial pressure re-equilibrated over roughly one hour and data from these periods were omitted from analyses."*

Full citation: Pereira, E., U. Schuster, V. Rérolle, P. Brown, T. Gkritzalis, B. Downing, K. Simpson, C. Lønborg, G. Carlin, S. Aßmann, and R. Spaulding, 2017: Biogeochemical Parameters: pCO2. In: A user's guide for selected autonomous biogeochemical sensors. An outcome from the 1st IOCCP International Sensors Summer Course [Lorenzoni, L., M. Telszewski, H. Benway, A. P. Palacz (eds.)]. IOCCP Report No. 2/2017, pp. 31-42.

[revised manuscript text omitted]